# Composition change-driven texturing and doping in solution-processed SnSe thermoelectric thin films

Seung Hwae Heo[1], Seungki Jo[1], Hyo Seok Kim[2], Garam Choi[2], Jae Yong Song[3], Jun-Yun Kang[4], No-Jin Park[5], Hyeong Woo Ban[1], Fredrick Kim[1], Hyewon Jeong[1], Jaemin Jung[6], Jaeyoung Jang [6], Won Bo Lee[2], Hosun Shin[3] & Jae Sung Son[1]

The discovery of SnSe single crystals with record high thermoelectric efficiency along the *b*-axis has led to the search for ways to synthesize polycrystalline SnSe with similar efficiencies. However, due to weak texturing and difficulties in doping, such high thermoelectric efficiencies have not been realized in polycrystals or thin films. Here, we show that highly textured and hole doped SnSe thin films with thermoelectric power factors at the single crystal level can be prepared by solution process. Purification step in the synthetic process produced a SnSe-based chalcogenidometallate precursor, which decomposes to form the SnSe$_2$ phase. We show that the strong textures of the thin films in the *b*–*c* plane originate from the transition of two dimensional SnSe$_2$ to SnSe. This composition change-driven transition offers wide control over composition and doping of the thin films. Our optimum SnSe thin films exhibit a thermoelectric power factor of 4.27 $\mu$W cm$^{-1}$ K$^{-2}$.

[1] School of Materials Science and Engineering, Ulsan National Institute of Science and Technology (UNIST), Ulsan 44919, Republic of Korea. [2] School of Chemical and Biological Engineering, Institute of Chemical Processes, Seoul National University, Seoul 08826, Republic of Korea. [3] Center for Convergence Property Measurement, Korea Research Institute of Standards & Science (KRISS), Daejeon 34113, Republic of Korea. [4] Korea Institute of Materials Science, 797 Changwon-daero, Seongsan-gu, Changwon, Gyeongnam 51508, Republic of Korea. [5] School of Advanced Materials Science and Engineering, Kumoh National Institute of Technology, Gumi, Gyeongbuk 39177, Republic of Korea. [6] Department of Energy Engineering, Hanyang University, Seoul 04763, Republic of Korea. These authors contributed equally: Seung Hwae Heo, Seungki Jo. Correspondence and requests for materials should be addressed to H.S. (email: hshin@kriss.re.kr) or to J.S.S. (email: jsson@unist.ac.kr)

Thermoelectric (TE) materials are popular because they enable energy conversion between heat and electricity, offering diverse applications such as power generation from waste heat and environmentally friendly refrigeration[1,2]. The energy conversion efficiency of a TE device, which generally depends on the inherent properties of the TE materials, is dictated by the dimensionless figure-of-merit: $ZT = (S^2 \sigma T/\kappa)$, where $S$, $\sigma$, $\kappa$, and $T$ are the Seebeck coefficient, electrical conductivity, thermal conductivity, and absolute temperature, respectively. Currently, the workhorse TE materials include compound semiconductors of chalcogenides, clathrates, skutterudites, and half-Heusler alloys, and strategies to tailor their structural and electronic characteristics to achieve high ZT values have been widely explored. One such strategy is nanostructuring of the TE materials, where formation of grain boundaries or nano-precipitates give rise to a number of interfaces that reduce the thermal conductivity of the materials[3–10]. Another strategy involves band structure engineering, whereby concepts of energy filtering, convergence of electronic bands and the distortion in density of states (DOS) are used to enhance the power factor ($S^2\sigma$)[11,12]. Unfortunately, these strategies generally require a tight control over the forming stage of the materials, negatively affecting the cost structure in manufacturing[13].

Recently, it was reported that SnSe single crystals inherently have ultralow thermal conductivity and consequently have ZT values higher than 2.3 in the $b$–$c$ plane[14]. SnSe is a compound semiconductor composed of earth-abundant elements and has an indirect band gap of 0.86 eV. It adopts a layered orthorhombic structure with the $Pnma$ space group at room temperature and the individual layer consists of double atomic layers of distorted rocksalt slabs. This anisotropic crystal structure is directly reflected by anisotropic transport properties, where the electrical and thermal conductivity in the $b$–$c$ plane is an order of magnitude higher than those along the $a$-axis. At ~800 K, a phase transition from $Pnma$ to a high symmetry $Cmcm$ space group occurs[14]. This transition, which decreases the band gap and increases carrier concentration, improves the electrical properties of SnSe, which, in turn, results in an exceptionally high ZT of 2.6 at 923 K along the $b$-axis. However, SnSe single crystals have faced two major challenges due to these structural characteristics. First, this type of crystallographic feature has poor mechanical properties, which limits the practical applications of this material[15]. Second, at temperatures below ~800 K, SnSe single crystals with the $Pnma$ space group have low carrier concentrations leading to poor TE properties.

Accordingly, significant efforts have been devoted to improving the mechanical properties by preparing SnSe polycrystals using hot-pressing[15–19] or spark plasma sintering[20–26], however, these polycrystals typically have ZT values below 1. The low ZT values are likely due to the anisotropic TE properties of SnSe single crystals (low ZT of 0.8 along the $a$-axis)[15–26]. One way to improve ZT values is by maximizing the texturing in SnSe polycrystals during the preparation stage, where the typical sintering process would not fulfil this purpose in bulk-scale materials[16,20–22,26,27]. To improve TE properties at temperatures below ~800 K, different dopants for SnSe have been explored. For example, Na doped SnSe single crystal was reported to exhibit ZT ranging from 0.7 to 2.0 at 300 K to 773 K[28]. However, doping is less effective for SnSe polycrystals because polycrystals have less texturing and form defects easily, which can degrade the electrical properties[15,17–19,23–25,29–31].

Thin film fabrication offers tremendous scope for ZT enhancement of SnSe polycrystal within the context of texturing and doping. Typically, the structural characteristics of materials in thin films are determined by interface and strain energy at the forming stage. This offers a wide range of ways to control texturing and doping in thin films through the choice of substrates, the coating process and the manipulation of post-treatment conditions. Furthermore, thin film thermoelectrics are potentially useful for a wide range of applications including micro-Peltier-coolers, micro-generators, and miniature sensors as well as flexible TE power supply for low-power consumption devices[32]. However, despite its importance, few studies on SnSe thin films have been reported[33–36].

Here, we report the high TE performance of hole doped and highly textured SnSe thin films fabricated by a low-cost and scalable solution process. The highest power factor of p-type SnSe thin films reached 4.27 µWcm$^{-1}$ K$^{-2}$ at 550 K, higher than reported values of SnSe single crystals. Such high TE performances of the solution-processed SnSe thin film were achieved by the synergistic effect of textured microstructure, controlled compositions, and film continuity, which were realized through the non-hydrazine synthesis of molecular SnSe-based chalco-genidometallate (ChaM) and the controlled transition from a two-dimensional (2D) transition metal dichalcogenide SnSe$_2$ phase to a SnSe phase.

## Result

**Synthesis of molecular SnSe ink solution.** The synthetic route for the SnSe inks involves two steps of dissolving of SnSe powder and the purification (Fig. 1a). We found that purifying the SnSe ink solution was important because it excluded undesired reactions and ensured film continuity and texturing. The initial SnSe solution was synthesized by the reported cosolvent approach in which SnSe powder was dissolved in mixed solvents of ethylene-diamine and ethanedithiol instead of widely used hydrazine solvent due to its high-level toxicity[37–41]. This chemical route generated the fully solubilized SnSe solution. The SnSe solution was further purified with anti-solvent of acetonitrile to precipitate out the SnSe ChaM. This precipitate was easily solubilized in various polar solvents such as $n$-methylformamide (NMF), dimethyl sulfoxide (DMSO), and ethylenediamine, among which ethylenediamine was chosen as carrier solvent for SnSe ink solution considering the ink processability (Supplementary Fig. 1).

This SnSe ink solution was deposited onto a hydrophilized glass by a multiple spin coating process with a short-time heat treatment to achieve the desired thickness of a film. The film was further annealed at 400 °C for an appropriate time. Scanning electron microscopy (SEM) and film photograph showed that unpurified SnSe solution did not form a continuous film when spin-coated on a glass substrate, and the SEM image shows the particulate morphology with multiple pinholes and pores in microstructures, which should hinder the charge carrier transport and eventually lower TE performance (Supplementary Fig. 2a,b). Furthermore, X-ray diffraction (XRD) pattern of this film corresponded to the Sn$_2$SSe reference (JCPDS 71–5582) rather than SnSe (Supplementary Fig. 3), indicating that the undesired reaction between Sn within SnSe compound and thiol from the solvents occurred upon heating[41]. In contrast, purified SnSe ink solution formed a perfect surface coverage on the substrate. SEM image shows the plate-like layered morphology with the grain sizes of several hundreds of nanometres (Fig. 1b). From the cross-sectional SEM image, the thickness appeared highly uniform and was estimated to be 85 nm for a film spin-coated four times (inset in Fig. 1b). The mirror-like reflection, typically observed in highly uniform and smooth films, further confirmed the SnSe thin films were also uniform at the cm-scale (Fig. 1a). More importantly, the XRD pattern of the SnSe thin film exclusively shows the $a$-axis peaks of (200), (400), and (800) with negligibly weak other peaks, (Fig. 1c) indicating the highly oriented and textured grains in the $b$–$c$ plane. To estimate the degree of orientation, we calculated an

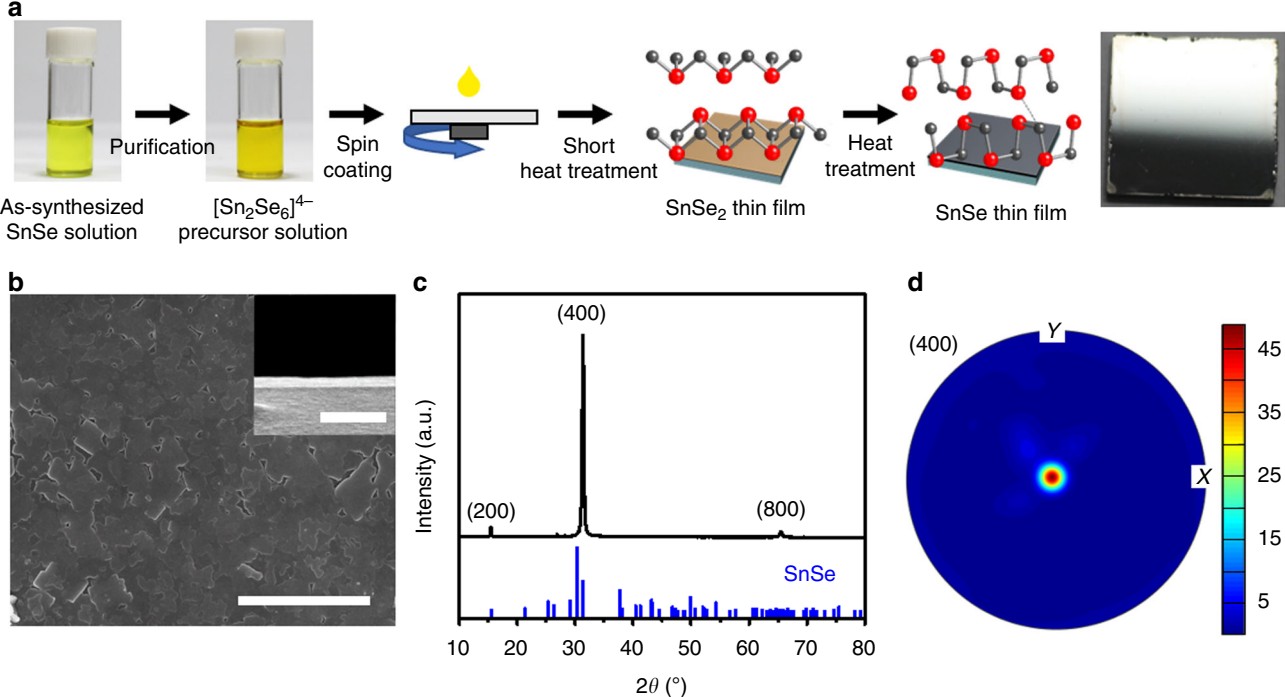

**Fig. 1** Solution-processed fabrication of highly textured SnSe thin films. **a** Schematic illustration of the fabrication process of SnSe thin films. SnSe ink solution was synthesized by a two-step process, which involves dissolving SnSe powder in a solvent composed of ethylenediamine and ethanedithiol, and purifying the resulting solution with acetonitrile to obtain a precipitate. The precipitate was dissolved in ethylenediamine to obtain the SnSe ink solution. The ink was spin-coated on a glass substrate and the film was heated for 1 min at 400 °C. The coating process was repeated to obtain the desired thickness. Finally, the film was annealed at 400 °C for an appropriate time. The photograph shows the fabricated film had a mirror-like reflection, indicating high uniformity. **b** SEM image of the SnSe thin film fabricated with purified precursor solution shows full coverage of SnSe on the substrate. Scale bar, 5 μm. Cross-sectional SEM image (inset) shows a uniform thickness of 85 nm. Scale bar, 500 nm. **c** XRD pattern of the SnSe thin film fabricated with purified precursor solution clearly shows peaks indexed to the (200), (400), and (800) planes, indicating the highly oriented texture. Vertical lines indicate the pattern for orthorhombic SnSe reference (JCPDS 32–1382). **d** Pole figure of the (400) plane in the SnSe thin film fabricated with purified precursor solution. Colours of the strips indicate the pole density in multiples of random distribution. High intensity at the centre clearly demonstrates the texturing in the thin film

orientation factor $F$ of $(h00)$ plane using Lotgering's method[42]. $F$ is evaluated by following Eqs. (1)–(3),

$$F = \frac{p - p_0}{1 - p_0} \qquad (1)$$

$$p = \frac{\sum I(h00)}{\sum I(hkl)} \qquad (2)$$

$$p_0 = \frac{\sum I_0(h00)}{\sum I_0(hkl)} \qquad (3)$$

where $I$ and $I_0$ are the intensities of the XRD peaks for the sample and the reference. $p$ and $p_0$ represent the relative quantity of $(h00)$ reflections to that of all reflections for the sample and the reference, respectively. The estimated anisotropic factor of the current SnSe thin film was as high as 0.89, which is close to that of reported SnSe crystal synthesized by the zone-melting process[21]. This texturing was further demonstrated by the strong intensity at the centre of the (400) pole figure for the SnSe thin film (Fig. 1d). Moreover, the inverse pole figure plot in normal direction to the film shows a very high intensity only at the [100] direction while there were no remarkable peaks in the horizontal and vertical directions of the plot. (Supplementary Fig. 4) These results clearly demonstrate strong texture development by which the film normal direction is mostly parallel to crystalline $a$ direction.

To understand the effect of purification, we investigated the SnSe solution before and after purification using inductive coupled plasma optical emission spectroscopy (ICP-OES), UV-Vis absorption and Raman spectroscopy. We found that the Sn:Se atomic ratio, estimated from ICP-OES analysis, changed from 1:1 to 1:~2.6 after purification. The composition of this purified solution is similar to the well-defined ChaM compounds, $Sn_2Se_6^{4-}$ or $Sn_4Se_{10}^{4-}$ [43,44]. Furthermore, the purified solution showed an absorption peak at 350 nm in the UV-Vis spectrum, which matches that of $Sn_2Se_6^{4-}$ ChaM[45]. Meanwhile, a broad absorption band at 300–400 nm was observed in the UV-Vis spectrum of the unpurified solution (Fig. 2a). In the Raman analysis, the unpurified solution shows a broad spectrum with peaks at 200 and 280 cm$^{-1}$, which do not match the vibration frequency of any SnSe-based ChaMs (Fig. 2b). On the other hand, distinctive peaks at 180 and 196 cm$^{-1}$ for the purified solution can be indexed to vibrational modes of the adamantanoid $Sn_4Se_{10}^{4-}$ and double-centred $Sn_2Se_6^{4-}$ ChaMs, respectively[46,47]. The broad peaks at ~250 cm$^{-1}$ may originate from the overlapping of multiple peaks coming from the ChaMs, which are expected to have vibrational modes ranging from 220 to 280 cm$^{-1}$ (Fig. 2b). These results suggest that the purified SnSe precursor solution likely contained $Sn_2Se_6^{4-}$ ChaM along with a small portion of $Sn_4Se_{10}^{4-}$ ChaM and counter cations from ethylenediammonium $(C_2H_4(NH_3)_2^{2+})$, while left over Sn ions are discarded along supernatant. This molecular SnSe precursor solution ensured high uniformity in the thin film by excluding undesired

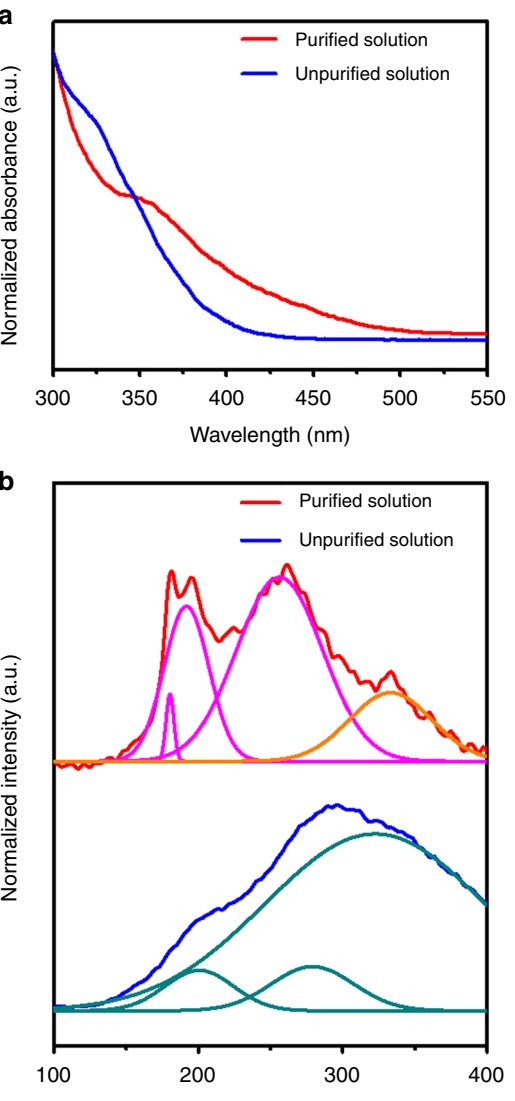

**Fig. 2** Synthesis of the SnSe ChaM precursor. **a** UV-Vis absorption spectra of SnSe precursors before and after the purification. The distinctive peak at 350 nm for the purified solution matches the reported absorption band of $Sn_2Se_6^{4-}$ ChaM. **b** Raman spectra of SnSe precursors before and after the purification. While broad and unidentifiable peaks were observed with unpurified precursor, after purification, several clear peaks were emerged at 191 and 257 cm$^{-1}$, and at 180 cm$^{-1}$, which are well matched with the vibration modes of the reported $Sn_2Se_6^{4-}$ and $Sn_4Se_{10}^{4-}$ ChaMs, respectively. Pink, orange, and turquoise curves represent deconvolution of Raman spectra

reactions and evaporations related to impurities at the forming stage.

**Oriented texture by controlled transition from SnSe$_2$ to SnSe.** Understanding the mechanism for texturing in SnSe polycrystal is critical for the development of highly efficient TE materials with structural anisotropy. To investigate the mechanism of texturing in the current SnSe thin films, we monitored the structural evolution of the as-coated thin film by XRD, SEM, and Raman spectroscopy during heat treatment. We found a preferential formation of a SnSe$_2$ phase as an intermediate structure. SnSe$_2$ is a well-known transition metal dichalcogenide with a 2D-layered structure. Due to the significant differences in the surface free

energies between crystallographic planes, such 2D materials energetically form highly oriented structures in thin films[48]. Here, we also observed the formation of highly textured SnSe$_2$ phase in the thin film at 300 °C. The XRD pattern of the thin film showed only the $c$-axis peaks of (001), (003), and (005), agreeing with the typical XRD patterns of textured 2D materials (Fig. 3a). At 350 °C, while the pattern for SnSe$_2$ remained, the peaks to be indexed as the textured SnSe phase started to appear in the XRD patterns. At 400 °C, the SnSe$_2$ phase peaks had completely disappeared, and the pattern of the highly textured SnSe structure in the $b$–$c$ plane was solely observed. The microstructural change during the transition was further investigated by SEM. At 300 °C, the thin film appeared smooth and uniform in microstructure (Fig. 3b). At 350 °C, the plate-like structure, which is expected to be SnSe, began to appear (Fig. 3c). This result is consistent with the XRD analysis where the SnSe phase first appeared at 350 °C. Finally, highly textured grains were observed at 400 °C (Fig. 3d). These grains were larger than those seen in the SnSe$_2$ thin film at 300 °C, suggesting that the transition accompanied the lateral growth of the SnSe grains in the $b$–$c$ plane. The Raman spectrum of the thin films annealed at 300 °C exclusively showed two peaks at 117 and 185 cm$^{-1}$, corresponding to the vibration modes of $E_g$ and $A_{g1}$ of SnSe$_2$ phase, respectively[49]. On the contrary, at 400 °C, these peaks are totally disappeared and new peaks at 70, 127 and 150 cm$^{-1}$ and at 107 cm$^{-1}$ appeared in the Raman spectrum (Supplementary Fig. 5), which are exactly matched to the $B_{3g}$ and to the $A_g$ vibration modes of the SnSe phase[50]. These experimental observations demonstrated that the highly oriented texture in the current SnSe thin film originates from the transition of the pre-formed textured SnSe$_2$.

This transition from SnSe$_2$ to the Se-deficient phase indicate Se evaporation occurred upon heating. To understand this phenomenon, we characterized the SnSe thin films annealed at various temperatures by energy dispersive X-ray spectroscopy (EDS) analysis (Fig. 3e). The spectra show that the peak intensity for Se decreased with higher annealing temperatures. The composition ratio of Sn and Se, estimated by EDS analysis (Supplementary Table 1), ranged from 1:1.6 to 1:0.96 when annealing temperatures increased from 300 °C to 400 °C, revealing that the evaporation of Se led to the transition from SnSe$_2$ to SnSe in the thin film. These results agree with thermogravimetric analysis (TGA) of the current ChaM precursor (Supplementary Fig. 6). After slight decline of weight at ~140 °C by the evaporation of remnant solvent, TGA spectrum shows a sharp decline at ~187 °C, indicating a decomposition of the ChaM precursor that might be attributable to the formation of $Sn_2Se_6^{4-}$ dimers. At higher temperature of ~237 °C, another sharp weight loss can be observed, which is a point where excess Se is evaporated, and forming SnSe$_2$ crystal structure. The total weight loss occurred from these two decomposition steps is 34.3% (from 154 to 300 °C), which agrees with the calculated value of 33.8% from $Sn_2Se_6^{4-}$ with ethylenediammonium as the counter cation. The weight slowly, but continuously decreases after 300 °C, indicating the evaporation of Se from the SnSe$_2$ phase to form SnSe crystal eventually.

To further understand this transition thermodynamically, we conducted the first-principle calculation for Helmholtz free energy of both SnSe$_2$ and SnSe from the density functional theory (DFT). The calculations were performed with Projector Augmented Wave (PAW) method[51], Generalized Gradient Approximation (GGA-PBE) exchange-correlation functional[52] and DFT-D3 method[53] for van der Waals interactions, which are implemented in Vienna ab initio simulation package (VASP)[54]. The basic unit cell structures for calculation were shown in Fig. 4a (SnSe$_2$) and Fig. 4b (SnSe). As shown in the calculated Helmholtz free energy of SnSe$_2$ and SnSe (Fig. 4c),

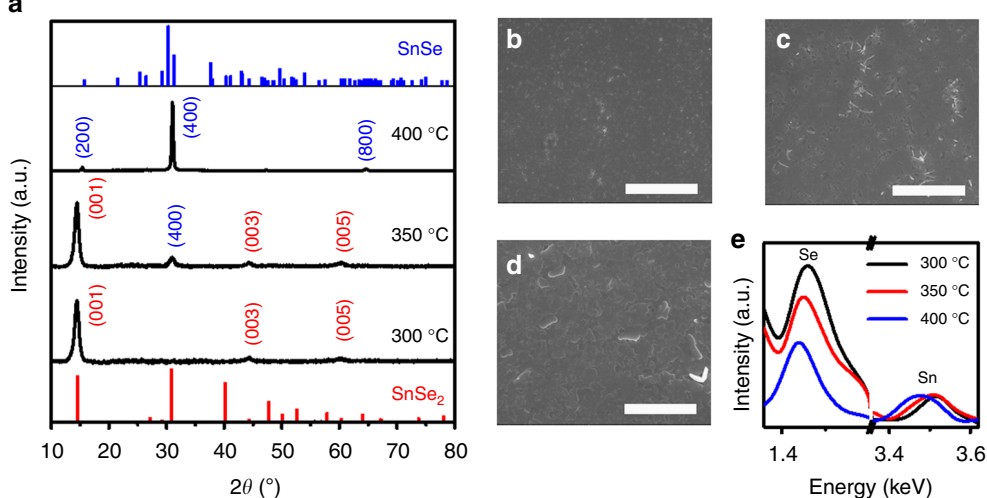

**Fig. 3** Composition change-driven texturing. **a** XRD patterns of the SnSe thin films upon heating. At 300 °C, only (00*l*) peaks of SnSe$_2$ were observed in the XRD pattern. At higher temperatures, these peaks progressively disappeared and the (*h*00) peaks of SnSe were exclusively detected at 400 °C. Vertical lines in panel **a** indicate the orthorhombic SnSe (blue, JCPDS 32–1382) and the hexagonal SnSe$_2$ phase (red, JCPDS 89–2939). **b–d** SEM images of the SnSe thin films at **b** 300 °C, **c** 350 °C, and **d** 400 °C. show progressive microstructural changes during the transition from SnSe$_2$ to SnSe. Scale bar, 5 µm. **e** EDS spectra of the SnSe thin films upon heating shows the decreasing Se content, compared with the normalized Sn content

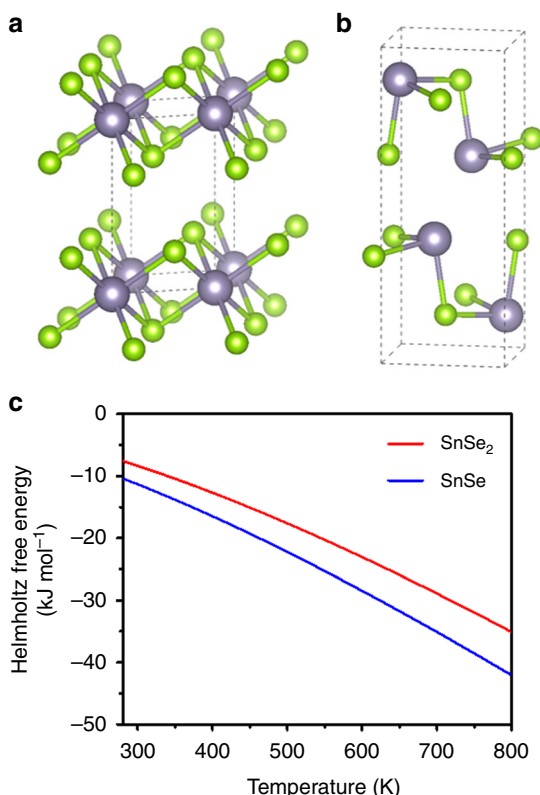

**Fig. 4** Thermodynamic free energy calculation. The unit cell structures for calculation. Supercells for **a** SnSe$_2$ and **b** SnSe were expanded as much as 4 × 3 × 2 and 2 × 2 × 2, respectively. **c** Helmholtz free energy of SnSe$_2$ and SnSe from first-principle calculation, where SnSe is thermodynamically more stable compared to SnSe$_2$ in the entire temperature range from 300 to 800 K

SnSe is thermodynamically more stable compared to SnSe$_2$ in the entire temperature range from 300 to 800 K. Accordingly, SnSe$_2$ can be predominantly synthesized only in certain circumstance, which is the Se excess condition. Since the Sn:Se atomic ratio of our purified SnSe ink solution was 1:~2.6, SnSe$_2$ can be formed as

intermediate state, then SnSe$_2$ was transitioned to SnSe as the heat treatment procedure processed and Se gradually evaporated. These theoretical results further support the transition from SnSe$_2$ to SnSe during heat treatment in the current study.

**Hole doping by compositional engineering**. The controlled doping in TE materials is crucial to ensure the optimum carrier concentration to maximize ZT values. Although intrinsic doping in SnSe remain poorly understood, thanks to direct observation of Sn vacancies (V$_{sn}$) in a SnSe crystal and the theoretical study predicted that the formation of V$_{sn}$ is more favourable than other defects, it is widely accepted that intrinsically formed Sn vacancies (V$_{sn}$) is a key factor for adjusting the hole concentration[55–58]. The observed transition from SnSe$_2$ to SnSe upon heating offers an additional degree of control over the phase and composition of SnSe thin films. Accordingly, we systematically control the heat treatment time to optimize composition and further investigate the hole doping effect.

Accordingly, we annealed the deposited SnSe thin films at 400 °C for 1, 5, 9, and 13 min each, and monitored the change in composition at these different heating times using EDS. The compositions of these films were identified as SnSe$_{1.31}$, SnSe$_{1.16}$, SnSe$_{1.01}$, and SnSe$_{0.89}$ for the 1, 5, 9, and 13 min annealing times, respectively (Supplementary Table 2). While heat treatment gradually decreased the Se content in the SnSe thin films, noticeable change was not observed in the grain structure; SEM images show every sample maintained the plate-like layered morphology (Supplementary Fig. 7). In contrast, when the thin film was annealed longer than 30 min, the plate-like textured microstructures were partially deformed while the perpendicularly oriented large-sized plate were newly appeared (Supplementary Fig. 8a). The elemental mapping image of this sample (Supplementary Fig. 8b) reveals that Sn and Se are homogeneously distributed over the entire area without the local accumulation, indicating the composition of large-sized plates is identical to that of the matrix. However, the composition of this sample estimated by EDS analysis is only 1:0.81 of the atomic ratio of Sn:Se, where the Se content is much lower than those observed in other samples (1–13 min heat-treated). In fact, such a Se deficiency critically affected the electrical properties; e.g. the

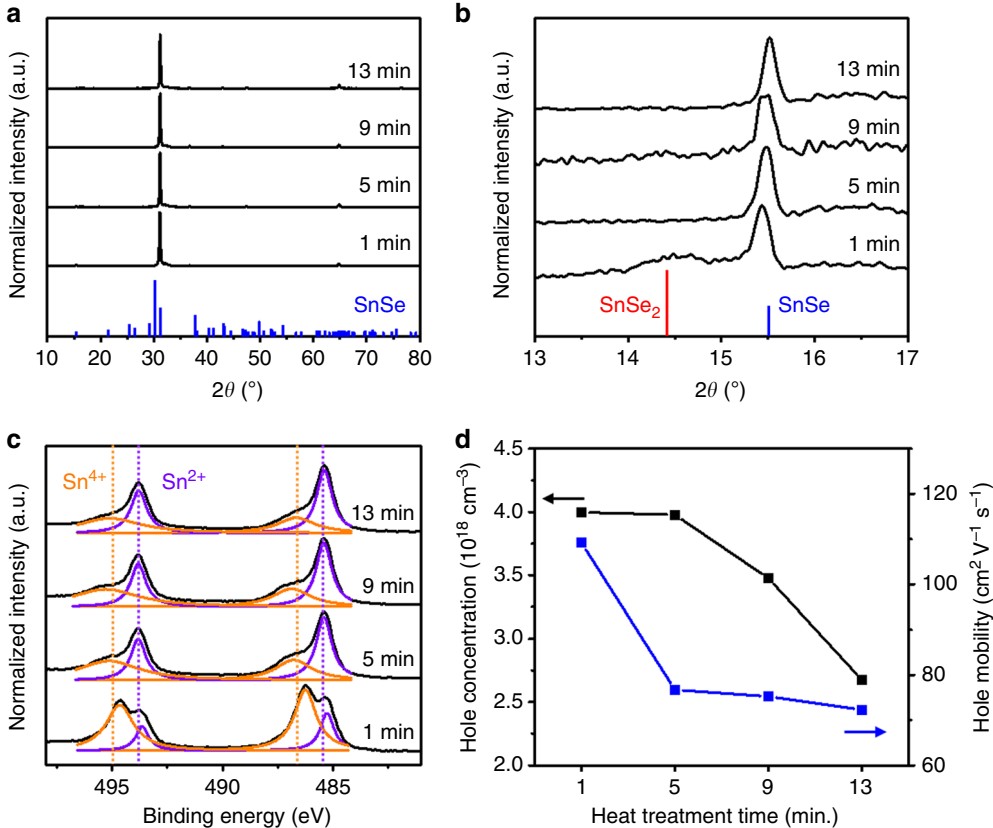

**Fig. 5** Compositional engineering of SnSe thin films for hole doping. **a–c** Structural and compositional characterization of SnSe thin films annealed at 400 °C for 1, 5, 9, and 13 min by **a**, **b** XRD and **c** XPS analyses. Vertical lines in **a** and **b** indicate the orthorhombic SnSe (blue, JCPDS 32–1382) and the hexagonal $SnSe_2$ phase (red, JCPDS 89–2939), respectively. Regardless of annealing time, all thin films showed XRD patterns that corresponded to the SnSe phase. Particularly, the enlarged XRD patterns in **b** for $2\theta$ ranging from 13° to 17° show the residual $SnSe_2$ phase in thin films annealed for 1 min at 400 °C. XPS spectra of thin films in the region for $3d_{5/2}$ and $3d_{3/2}$ of Sn show distinctive peaks at 485.4 and 493.8 eV (purple dotted lines), which are identified as the Sn(2+) oxidation state. Additional peaks at 486.3 and 494.7 eV (orange dotted lines) detected in the 1 min-treated sample can be indexed to the Sn(4+) oxidation state. **d** Graph shows hole mobilities and hole concentrations of SnSe thin films annealed at 400 °C for 1, 5, 9, and 13 min obtained by Hall effect measurement at room temperature

electrical conductivity of this sample is at least an order of magnitude lower than other samples.

The highly textured features in the XRD patterns of the SnSe thin films heated at different times were seen throughout whole samples (Fig. 5a). Interestingly, a broad and subtle $SnSe_2$ (001) peak was detected in the sample heated for 1 min; this peak likely came from a trace of residual $SnSe_2$ phase (Fig. 5b). The existence of $SnSe_2$ phase without severe deformation of SnSe grain structure and peak shift related to alloying in the XRD pattern may suggest that the $SnSe_2$ phase exist as an embedded structure within the SnSe crystal. Furthermore, the X-ray photoelectron spectroscopy (XPS) spectrum displayed the $3d_{5/2}$ and $3d_{3/2}$ peaks of Sn (Fig. 5c). Throughout all samples at different heating times, sharp peaks at 485.4 eV and 493.8 eV remain almost identical. These peaks, which were identified as the Sn(2+) oxidation state, showed that the SnSe phase is well-defined in the fabricated thin films. Moreover, distinctive peaks at 486.3 and 494.7 eV representing the Sn(4+) oxidation state were seen in the 1 min-heated sample, demonstrating that the remnant $SnSe_2$ phase rapidly transitions to the SnSe phase with heat treatment[33,59].

Hole-doping effect by composition engineering were investigated by Hall effect measurement on SnSe thin films at room temperature (Fig. 5d). With increasing heat treatment time, the hole concentration progressively decreased from $4.0 \times 10^{18}$ cm$^{-3}$ for the 1-min-heated sample to $2.7 \times 10^{18}$ cm$^{-3}$ for the 13-min-

heated sample. Moreover, these hole concentrations are an order of magnitude higher than that of single crystalline SnSe and were similar to the reported value for spark plasma sintered $Sn_{1-x}$ Se polycrystal that exhibit peak ZT value of 2.1, indicating the hole doping effect of SnSe thin films by compositional engineering[60]. Meanwhile, despite the largely different Se content, 1- and 5-min-heated samples exhibited nearly identical hole concentrations. This could be explained by the limited concentration of the $V_{Sn}$ formed by Se precipitation in the SnSe phase. Wei and co-workers observed a similar change in hole concentration for composition engineered SnSe polycrystals[61]. On the other hand, it should be noted that the highest hole mobility of 109 cm$^2$ V$^{-1}$ s$^{-1}$ achieved by the 1-min-heated sample is higher than those reported for SnSe polycrystals. This shows that our SnSe thin films have excellent structural quality. With increasing heat treatment time, the hole mobility further decreased to 72.3 cm$^2$ V$^{-1}$ s$^{-1}$ for the 13-min-heated sample. This behaviour might be attributed to the formation of Se-deficient defects, such as Se vacancies arising from the evaporation of Se during heat treatment, because these point defects can scatter hole carriers. At the same time, we cannot exclude the possibility that the residual $SnSe_2$ phase contributed to the overall electrical properties in less-heat-treated thin films. Recently, Wang et al. reported that locally embedded $SnSe_2$ microdomains in the SnSe phase lead to the hole doping in SnSe through the interfacial charge transfer[62]. Likewise, the

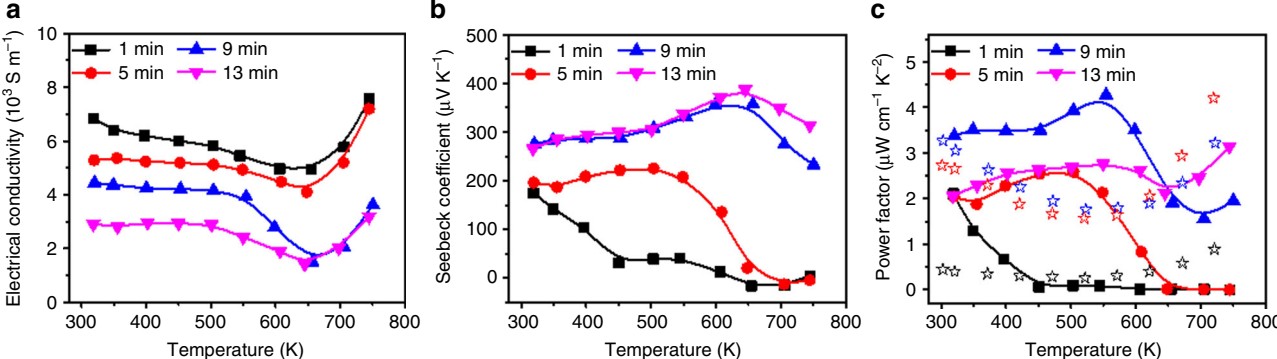

**Fig. 6** Temperature-dependent TE properties of SnSe thin films. **a–c** Graphs show electrical conductivities (**a**), Seebeck coefficients (**b**), and power factors (**c**) for SnSe thin films annealed at 400 °C for 1, 5, 9, and 13 min. Stars in **c** indicate data points obtained with SnSe single crystals in *a* (black), *b* (red), and *c* (blue) axes[14]

electrical properties of the 1-min-treated SnSe sample might also be affected by the residual $SnSe_2$ phase existing as microdomains within the SnSe phase.

The texturing, hole doping, and continuous microstructure achieved in the SnSe thin films were clearly reflected by the excellent TE properties (Fig. 6). Temperature-dependent TE properties were measured at temperatures ranging from room temperature to 750 K. The electrical conductivities of all samples exhibited a negative temperature dependence in temperature below ~650 K (Fig. 6a), which was then changed to semiconducting behaviour for the higher temperature region, agreeing with the behaviour observed in SnSe single crystals with the *Pnma* space group[14]. Owing to the relationship of hole concentrations and hole mobility with heat treatment, the electrical conductivity decreased with increasing heat treatment time. The highest electrical conductivity of $7.6 \times 10^3$ S $cm^{-1}$ at 750 K was achieved by the 1-min-heated sample; this value is similar to or even higher than those reported for SnSe single crystals and polycrystals. The Seebeck coefficients of the samples were between 200 and 300 μV $K^{-1}$ at room temperature but increased with heat treatment time because Seebeck coefficients are inversely proportional to carrier concentrations. The 1- and 5-min-treated samples showed significant decrease in Seebeck coefficients as temperatures increased. This temperature dependence can be attributed to the residual $SnSe_2$ phase in less-heat-treated samples. Typically, 2D $SnSe_2$ crystals are reported to exhibit n-type semiconductor properties[44], in which the thermally excited minority carrier concentration can be increased at high temperature and it significantly suppress the Seebeck coefficient in the current p-type SnSe thin films. On the other hand, 9- and 13-min-treated samples showed similar behaviour to typical SnSe single crystals and polycrystals. The highest power factors achieved by the 9-min sample at 550 K was 4.27 μW $cm^{-1}$ $K^{-2}$. To the best of our knowledge, this value is at least an order of magnitude higher than the most recently reported SnSe thin films[33] and even higher than those measured along the *b*-axis of SnSe single crystals in the mid-temperature range[14].

The thermal conductivities of the SnSe thin films were estimated by the first-principles DFT calculations because accurate measurements of in-plane thermal conductivity are technically difficult to be obtained by typical methods such as 3Ω method or time-domain thermoreflectance (TDTR). The basic unit cell structure for the calculation was $Sn_{16}Se_{16}$ (Supplementary Fig. 9a). Moreover, considering the composition of our samples, we further calculated the thermal conductivities of $Sn_{15}Se_{16}$ and $Sn_{16}Se_{15}$ structures by Debye–Callaway model with the consideration of phonon scattering effect from Umklapp and normal processes, and point defect scattering. These $Sn_{15}Se_{16}$,

$Sn_{16}Se_{16}$, and $Sn_{16}Se_{15}$ are generally matched to the fabricated 5-, 9-, and 13-min-treated samples in the composition. The 1-min-treated sample was excluded for the calculation due to the distinct residual $SnSe_2$, causing a large deformation in the simulated unit cell structure. As shown in the Supplementary Fig. 9d, the calculation result with unit cell having atomic ratio 16:16 has the highest thermal conductivity of 1.53 W $m^{-1}$ $K^{-1}$ at room temperature, while $Sn_{15}Se_{16}$ and $Sn_{16}Se_{15}$ shows the values of 0.82 and 0.89 W $m^{-1}$ $K^{-1}$, respectively. The lower thermal conductivities of $Sn_{15}Se_{16}$ and $Sn_{16}Se_{15}$ are attributed to Sn or Se vacancies acting as scattering centres. Moreover, all models showed the negative temperature dependence of thermal conductivities, agreeing with the typical trend of semiconductors.

With the calculated thermal conductivities of $Sn_{15}Se_{16}$, $Sn_{16}Se_{16}$, and $Sn_{16}Se_{15}$, we estimated the ZT values of 5-, 9- and 13-min-treated samples. The maximum ZT value of 0.58 was achieved by 13-min-treated sample at 750 K. To the best of our knowledge, this value is the highest among the reported of SnSe thin films and compete with those measured along the *b*-axis of SnSe single crystals in the corresponding temperature range (Supplementary Fig. 10). Also, considering the similar temperature dependence of electrical properties with single crystalline bulk SnSe, the power factors of our samples may be further increased after 750 K. Considering that thin films typically have much lower thermal conductivity than bulk materials owing to interface or surface scattering of phonons, the actual ZT value of the current SnSe thin films is expected to be higher than the calculated.

## Discussion

In summary, we showed that highly textured SnSe thin films exhibiting TE power factors at the single crystal level (Supplementary Fig. 11) can be easily prepared by solution process using SnSe-based ChaM precursor. The high structural quality and texturing of the thin films were achieved through a purification step in the synthetic process to produce the molecular $Sn_2Se_6^{4-}$ ChaMs. We show that these precursors firstly decompose to form the $SnSe_2$ phase before forming the SnSe phase through Se evaporation achieved by heating. This transition from 2D $SnSe_2$ to SnSe offers an extensive degree of engineering over the texturing and the doping of the SnSe thin films, leading to particularly high TE power factors of 4.27 μW $cm^{-1}$ $K^{-2}$. Furthermore, this composition change approach offers a unique mechanism to secure texturing in 2D materials with controlled compositions.

Another important payoff of the current technology is a cost-effectiveness of a solution process for high performance SnSe thin films. The current SnSe ink solution can be further applied to more practical solution process to fabricate applicable SnSe thick

films. For example, the spray coating with the SnSe ink was performed on a glass substrate (10 mm × 10 mm). The SEM images of the spray-coated film (Supplementary Fig. 12a) shows plate-like grain morphology without any voids or cracks, very similar to those of the spin-coated thin films. The thickness estimated in the cross-sectional SEM image is around 1 μm (Supplementary Fig. 12b), sufficiently thick enough to be applied to micro-sized thin film thermoelectric generators or micro-Peltier cooler[60]. Moreover, the XRD pattern of this sample (Supplementary Fig. 13a) exclusively shows the strong peaks indexed as (200), (400), and (800) planes of SnSe, demonstrating the texturing in the b–c plane. The estimated orientation factor F of (h00) plane using Lotgering's method is 0.75, very close to 0.89 of the spin-coated samples. More importantly, there is no secondary phases related to SnSe$_2$ in the XRD pattern, which indicates that the texturing mechanism driven by the composition change from SnSe$_2$ to SnSe works very well on a thick film fabricated by the spray coating process. The photograph of the thin film also show even surface (Supplementary Fig. 13b). These results clearly demonstrate the feasibility of the solution-processed fabrication of high performance SnSe thick films in the current study. We strongly believe that this method can be widely extended to cost-effective and scalable production of well-designed inorganic functional films.

## Methods

**Materials**. SnSe lump (99.999%) was purchased from Alfa Aesar. Ethanedithiol (≥98%), ethylenediamine (≥99.5%), acetonitrile (>99.8%), dimethyl sulfoxide (DMSO, >99.9%), and N-methylformamide (NMF, 99%) were purchased from Sigma Aldrich Chemical Co. All element and chemicals were used without further purification.

**Synthesis of SnSe solution**. The SnSe solution is synthesized by the reported cosolvent approach and the subsequent purification. Typically, 100 mg of ground SnSe powder was added in the mixture of 0.2 ml of ethanedithiol and 2 ml of ethylenediamine. After stirring for 2.5 h at 50 °C, SnSe powder completely dissolved to become pale yellow solution. To purify the synthesized SnSe precursor, the anti-solvents of acetonitrile were added into the SnSe solution in the anti-solvent:solvent volume ratios of 30:1. This mixture was centrifugated at 13,400 rpm for 5 min to precipitate out the precursor, which was subsequently dissolved in 2.2 ml of various solvents such as ethylenediamine, N-methyl formamide, and dimethyl sulfoxide. The whole synthesis processes were conducted in nitrogen-filled glove box.

**SnSe thin film fabrication**. Glass substrates were washed by using methanol, acetone, and isopropanol to remove the organic residues, followed by O$_2$ plasma treatment. A volume of 40 μl of the purified SnSe ink solution was spin-coated on the hydrophilized glass substrate by the spin-coating process at 2000 rpm for 30 s. The coated thin films were dried at 70 °C for at least 5 min to completely evaporate the residual ethylenediamine. This film was annealed for 1 min on the hot plate with the pre-set temperature of 400 °C. This annealed thin film was cooled at room temperature for 10 min, then same deposition process was repeated for several times to obtain the desired thickness. The final thin film was finally annealed at 400 °C for an appropriate time (1, 5, 9, and 13 min). The whole fabrication processes described above were conducted in nitrogen-filled glove box.

**SnSe spray coating**. The spray coating of SnSe was proceeded using a spray gun (HP-TH, IWATA) on the hydrophilized glass substrate (10 mm × 10 mm), which was placed on the hot plate with the pre-set temperature of 80 °C. 5 psi of pressure was applied to the spray gun, and the distance between the nozzle of spray gun and the glass substrate was 1.5–3.0 cm. The whole coating process described above was conducted in nitrogen-filled glove box.

**Helmholtz free energy calculation**. Density functional theory (DFT) calculations were performed with Projector Augmented Wave (PAW) method[49], Generalized Gradient Approximation (GGA-PBE) exchange-correlation functional[50], and DFT-D3 method[51] for van der Waals interactions, which are implemented in Vienna ab initio simulation package (VASP)[52]. Valence electrons were set to 4d, 5s, and 5p for Sn, 4s and 4p for Se. Fully relaxed crystal structures of SnSe$_2$ and SnSe are confirmed as P-3m1 and Pnma, respectively. The relaxed lattice constants are 3.858/3.858/6.205 (a/b/c) for SnSe$_2$ and 11.577/4.171/4.527 for SnSe, respectively. Free energies are post-processed by using phonopy[63] from DFT calculations of the relaxed structures. Helmholtz free energy (F) is calculated based on lattice

dynamics[64] and partition functions (Z) of phonon frequencies, where $F = -k_B T \ln Z$. Detail formulations are well described in developer's papers[63,65].

**Thermal conductivity calculation**. In Sn$_{16}$Se$_{16}$ case, the lattice thermal conductivity was computed from a solution of the linearized phonon Boltzmann transport equation (LBTE)[66]. In single-mode relaxation time (SMRT) approximation, the equation can be written in a closed form,

$$\kappa = \frac{1}{NV_0} \sum_\lambda C_\lambda \nu_\lambda \otimes \nu_\lambda \tau_\lambda^{SMRT} \tag{4}$$

where $V_0$ is the volume of a unit cell, and $\nu_\lambda$ and $\tau_\lambda^{SMRT}$ are the group velocity and SMRT of phonon mode $\lambda$, respectively.

In order to obtain phonon dispersion relation and phonon lifetimes, PHONOPY[63], and PHONO3PY[66] code were employed. Harmonic and anharmonic interatomic force constants (IFCs) were calculated using VASP[67,68] with the PBEsol[69] exchange–correlation functional. The atomic displacement was set to 0.03 Å, and the k-meshes for the force calculations were set to $2 \times 2 \times 2$. To calculate the lattice thermal conductivity, the q-meshes were set to $7 \times 7 \times 7$.

In Sn$_{16}$Se$_{15}$ and Sn$_{15}$Se$_{16}$ case, lattice thermal conductivity calculations were performed using the Callaway model, which was tuned to match with Boltzmann Transport Equation (BTE) results, suggested by Wu et al.[70]. Three scattering factors were considered: the intrinsic Umklapp and Normal processes, and point defect scattering. In the relaxation time approximation, the lattice thermal conductivity $\kappa_{latt}$ can be written as:

$$\kappa_{latt} = \frac{k_B}{2\pi^2 v} \left( \frac{k_B T}{\hbar} \right)^3 \int_0^{\frac{\theta_D}{T}} \frac{\tau_c z^4 e^z}{(e^z - 1)^2} dz \tag{5}$$

The total relaxation time ($\tau_c$) consists of individual scattering mechanism via Matthiessen's rule:

$$\tau_c^{-1} = \tau_U^{-1} + \tau_N^{-1} + \tau_v^{-1} \tag{6}$$

Umklapp process,

$$\tau_U^{-1} = \frac{\hbar \gamma^2}{Mv^2 \theta_D} \omega^2 T \exp\left( -\frac{\theta_D}{3T} \right) \tag{7}$$

Normal process,

$$\tau_N^{-1} \approx \beta \tau_U^{-1} \tag{8}$$

Vacancy scattering,

$$\tau_v^{-1} = \frac{\omega^4 \delta^3}{4\pi v^3} y(1-y) \left[ -\frac{M_v}{M} - 2 \right]^2 \tag{9}$$

where $k_B$ is the Boltzmann constant, $\hbar$ is the Plank constant, $v$ is the sound (phonon-group) velocity along each principle axis at the long wave limit, $\theta_D$ is the axial Debye temperature, and $z$ is defined as $\hbar\omega/k_B T$, $\gamma$ is Grüneisen parameter, $M$ is average molar mass of one atom; $\beta$ is a fitting parameter for normal process; $\delta$ is the volume of a host atom; $M_v$ is the molar mass of the missing atom (vacancy), $y$ is the molar ratio of Sn vacancies.

**Materials characterization**. X-ray photoelectron spectroscopy (XPS): The spectrum of the SnSe thin films was obtained using a K-alpha (ThermoFisher) with a Mg Kα X-ray monochromatic source.

X-ray diffraction (XRD): XRD patterns were collected by using D/MAX2500V/PC (Rigaku) with a Cu Kα X-ray source, which has a characteristic wavelength of 1.5418 Å.

Thermogravimetric (TGA) analysis: TGA analysis was conducted by using Q600 (TA instrument) with a heating rate of 5 °C min$^{-1}$ under the argon flow rate of 100 ml min$^{-1}$.

UV-Vis absorption spectroscopy: The absorption spectra were collected using a UV-2600 (Shimadzu) spectrophotometer.

Scanning electronic microscopy (SEM): The microstructure was characterized by using a field effect SEM (Nova-NanoSEM230, FEI and S-4800 Hitach high-Technologies) operated at 10 kV. The elemental analyses for SnSe thin films were conducted via energy dispersive X-ray spectroscopy (EDS) using Nova-NanoSEM230.

Inductive coupled plasma optical emission spectrometry (ICP-OES): The compositional analysis of the SnSe precursor was conducted using ICP-OES (700-ES, Varian).

Raman spectroscopy: The molecular structure analyses were conducted with Raman spectroscopy (Alpha300R, WITec and DXR, Thermo Fisher Scientific).

X-ray pole figure measurement: The pole figure was measured at an accelerating voltage of 40 kV and a current of 30 mA using an X-ray diffractometer (Bruker-

AXS D-5005 model) with Co-target (wavelength: 0.17902 nm) for (400) and (511). Orientation distribution function (ODF) was calculated using the experimental pole figures and MTEX tool box for MATLAB, from which the normalized pole figures and inverse pole figures were obtained.

**TE properties measurement**. The temperature-dependent TE properties were measured using a bulk/thin film thermoelectric measurement system (LSR3, Linseis Inc.), applying temperature difference between two points by 2–6 °C. To prevent the oxidation of the films during measurement, the measurement was conducted under $H_2/Ar$ atmosphere. The thermal stability of the samples during the measurement was confirmed by the repetitive measurement under thermal cycles. The Hall effect measurement at room temperature was conducted by a Hall measurement system (HMS-5300, ECOPIA), and mobility was calculated with electrical conductivities measured with Van der Pauw method.

## Data availability

The data that support the plots within this paper and other findings of this study are available from the corresponding author upon reasonable request.

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

## Acknowledgements

This research was supported by the Low-dimensional Materials Genome Development by Korea Research Institute of Standards and Science (KRISS−2016−16011070), Nano·Material Technology Development Program (NRF-2018M3A7B8060697) through the National Research Foundation of Korea (NRF) funded by the Ministry of Science and ICT (MSIT), and Basic Science Research Program (2015R1C1A1A01053599) through the NRF funded by the Ministry of Education.

## Author contributions

S.H.H., S.J., H.S., J.Y.S., and J.S.S. designed the experiments, analysed the data and wrote the paper. S.H.H., S.J., H.W.B., F.K., H.J., J. Jung, and J. Jang carried out the synthesis and basic characterization of the materials. H.S. performed the characterization of the thermoelectric properties. H.S.K., G.C., and W.B.L. carried out the first-principle calculation of materials. J.-Y.K. and N.-J.P. carried out the pole figure measurement of the materials. All authors discussed the results and commented on the manuscript.
