## [Peer Review File · Nature Communications]

Reviewers' comments:

Reviewer #1 (Remarks to the Author):

The central claim of this paper is that the authors have demonstrated a route to fabrication of highly-oriented thin films of SnSe that provides outstanding thermoelectric performance in the plane of the film, comparable to that of single crystals of the anisotropic SnSe phase. The specific orientation of the highly-textured film is responsible for the impressive properties. To the best of my knowledge, the work is original. The conclusions are well-justified by the data presented. The proposed mechanisms for formation of the highly-textured SnSe film, by phase transition from a highly-textured SnSe₂ film that, in turn, forms from well-defined molecular cluster precursors is also well supported. The demonstration of the ability to control carrier concentration and electrical conductivity via annealing time and temperature to optimize thermoelectric power factor is also important, and is clearly demonstrated

The language would benefit from further polishing, but is quite understandable, with only minor grammatical errors.

Although some interesting chemical insights are presented, the key innovation in this work lies in the film formation process, rather than more fundamental materials science or physics insights. Thus, I think this paper can only have the high impact expected for Nature Communications (strongly influencing thinking in the field) if the method actually provides a pathway to cost-effective practical formation of thick enough films to be useful in a device. The spin-coating approach, while the natural approach to use in a laboratory study, does not directly provide that pathway. Thus, if the texturing is dependent on depositing material ~ 20 nm at a time, with drying and annealing between deposition passes, then the process will not be very practical or cost-effective. Many of the advantages of solution-phase processing will be lost. If the authors could demonstrate that a similar degree of texturing can be achieved by bar-coating, doctor-blading, spray coating, or (even better) a flexographic or similar printing approach, which would deposit much thicker films per pass and which are inherently much more scalable than spin-coating, I think the impact of the paper would be greatly increased. Even if not, the paper is certainly worthy of publication, just perhaps not in a Nature family journal.

Sincerely, Mark Swihart, The University at Buffalo (SUNY)

Reviewer #3 (Remarks to the Author):

This paper "Phase transition-induced texturing and doping in solution-processed SnSe thermoelectric thin films" by Heo et al. reports the high thermoelectric performance of solution-processed SnSe thin film. The detailed synthetic chemistry for ink solution was shown to obtain molecular Sn₂Se₆ chalcogenidometallate complex. The most interesting part of this paper is the phase transition from SnSe₂ to SnSe and the resulting texturing in thin films, which also seems to be beneficial to control the composition and the related doping. The achieved power factor of $\sim 3 \mu\text{W cm}^{-1} \text{K}^{-2}$ by SnSe thin films is a really high value in thin film thermoelectrics and comparable to bulk SnSe. Although the strategies of texturing and doping in bulk SnSe have been studied in several papers to enhance the thermoelectric conversion efficiency, I think that the phase-transition approach for this material is new and can offer an important scientific insight to thermoelectric society. Moreover, the paper is well organized and the results and discussion including experimental and theoretical considerations is reasonable. In these regards, I recommend the publication of this paper in Nature Communications after addressing some minor issues, as presented below.

1. In XPS and XRD analyses of SnSe thin films, the substantial portion of SnSe₂ phase was detected in 1 min sample. I think this residual phase can significantly affect the carrier density and

mobility. For example, the authors already discussed n-type properties of SnSe₂ phase, affecting the thermoelectric properties of SnSe thin films in the manuscript. The authors need to discuss this point.

2. In 30 min sample, the author commented "the continuous microstructures deformed severely." However, the microstructure seen in SEM image is not so different from other samples and a new phase of rod structure was observed. I recommend the authors provide more detailed information of this sample.

Response to the reviewers' comments

The followings are the responses to the reviewers' comments for the manuscript "Composition change-driven texturing and doping in solution-processed SnSe thermoelectric thin films"

▪ Reviewer #1

General comment: The central claim of this paper is that the authors have demonstrated a route to fabrication of highly-oriented thin films of SnSe that provides outstanding thermoelectric performance in the plane of the film, comparable to that of single crystals of the anisotropic SnSe phase. The specific orientation of the highly-textured film is responsible for the impressive properties. To the best of my knowledge, the work is original. The conclusions are well-justified by the data presented. The proposed mechanisms for formation of the highly-textured SnSe film, by phase transition from a highly-textured SnSe₂ film that, in turn, forms from well-defined molecular cluster precursors is also well supported. The demonstration of the ability to control carrier concentration and electrical conductivity via annealing time and temperature to optimize thermoelectric power factor is also important, and is clearly demonstrated

Comment 1: The language would benefit from further polishing, but is quite understandable, with only minor grammatical errors.

Response: We have tried to our best to edit all grammatical errors by a native English speaker in the revised manuscript.

Comment 2: Although some interesting chemical insights are presented, the key innovation in this work lies in the film formation process, rather than more fundamental materials science or physics insights. Thus, I think this paper can only have the high impact expected for Nature Communications (strongly influencing thinking in the field) if the method actually provides a pathway to cost-effective practical formation of thick enough films to be useful in a device. The spin-coating approach, while the natural approach to use in a laboratory study, does not directly provide that pathway. Thus, if the texturing is dependent on depositing material ~20 nm at a time, with drying and annealing between deposition passes, then the process will not be very practical or cost-effective. Many of the advantages of solution-phase processing will be lost. If the authors could demonstrate that a similar degree of texturing can be achieved by bar-coating, doctor-blading, spray coating, or (even better) a

flexographic or similar printing approach, which would deposit much thicker films per pass and which are inherently much more scalable than spin-coating, I think the impact of the paper would be greatly increased. Even if not, the paper is certainly worthy of publication, just perhaps not in a Nature family journal.

Response: We agree with the reviewer that the scalable fabrication of high-quality SnSe thick films would be crucial to demonstrate the practicability of our technology. As the reviewer suggested, we have tried to conduct **the spray coating with our SnSe ink on a glass substrate (10 mm × 10 mm)** and characterize the texturing of the fabricated thin films by X-ray diffraction (XRD) and scanning electron microscopy (SEM) analyses. The SEM images of the spray-coated film (Fig. A1a and b) shows the plate-like grain morphology without any voids or cracks, very similar to those of the spin-coated thin films shown in the previous manuscript. **The thickness estimated in the cross-sectional SEM image is around 1 μm, sufficiently thick enough to be applied to micro-sized thin film thermoelectric generators or micro-Peltier cooler.** For example, Choi, et al. reported micro thermoelectric generator made of p-type polycrystalline Si, with thickness of 200-450 nm^{A1}. Moreover, **the XRD pattern of this sample (Fig. A2) exclusively shows the strong peaks indexed as (200), (400), and (800) planes of SnSe, demonstrating the texturing in the *b-c* plane.** The estimated orientation factor *F* of (*h*00) plane using Lotgering's method^{A2-A4} is 0.75, very close to 0.89 of the spin-coated samples. More importantly, there is no secondary phases related to SnSe₂ in the XRD pattern, which indicates that the texturing mechanism driven by the composition change from SnSe₂ to SnSe works very well on such a thick film fabricated by the spray coating process. These results clearly demonstrate that the feasibility of the solution-processed fabrication of high performance SnSe thin films using newly developed SnSe ink solution in the current study.

Fig. A1 | Microstructure of spray-coated SnSe thin film. **a**, SEM image of SnSe thin film fabricated via spray-coating. The sample shows the plate-like morphology in microstructure. Scale bar, 3 μm . **b**, Cross-sectional SEM image of same sample. Scale bar, 3 μm .

Fig. A2 | XRD pattern of SnSe thin film fabricated through spray coating method. XRD pattern show clear peaks indexed to the (200), (400), and (800) planes, confirmed its textured structure. Photograph of thin film (inset) show even surface of thin film.

We included the SEM images and XRD pattern of the spray-coated thin film (Fig. A1 and A2) in the revised Supplementary Information (Supplementary Fig. 12 and 13) and included the following sentences (page 16-17) and experimental details (page 18) in the revised manuscript.

“Another important payoff of the current technology is a cost-effectiveness of a solution process for high performance SnSe thin films. The current SnSe ink solution can be further applied to more practical solution process to fabricate applicable SnSe thick films. For example, the spray coating with the SnSe ink was performed on a glass substrate (10 mm × 10 mm). The SEM images of the spray-coated film (Supplementary Fig. 12a) shows the plate-like grain morphology without any voids or cracks, very similar to those of the spin-coated thin films. The thickness estimated in the cross-sectional SEM image is around 1 μm (Supplementary Fig. 12b), sufficiently thick enough to be applied to micro-sized thin film thermoelectric generators or micro-Peltier cooler⁶⁶. Moreover, the XRD pattern of this sample (Supplementary Fig. 13) exclusively shows the strong peaks indexed as (200), (400), and (800) planes of SnSe, demonstrating the texturing in the *b-c* plane. The estimated orientation factor *F* of (*h*00) plane using Lotgering’s method is 0.75, very close to 0.89 of the spin-coated samples. More importantly, there is no secondary phases related to SnSe₂ in the XRD pattern, which indicates that the texturing mechanism driven by the composition change from SnSe₂ to SnSe works very well on such a thick film fabricated by the spray coating process. These results clearly demonstrate that the feasibility of the solution-processed fabrication of high performance SnSe thick films in the current study. We strongly believe that this method can be widely extended to cost-effective and scalable production of well-designed inorganic functional films.”

“SnSe spray coating. The spray coating of SnSe was proceed using a spray gun (HP-TH, IWATA) on the hydrophilized glass substrate (10 mm × 10 mm), placed on the hot plate with the pre-set temperature of 80 °C. 5 psi of pressure was applied to spray gun, and distance from the nozzle of spray gun and the glass substrate was 1.5-3.0 cm. The whole coating process described above were conducted in nitrogen-filled glove box.”

References

- A1 Choi, J., Cho, K. & Kim, S. Flexible Thermoelectric Generators Composed of n-and p-Type Silicon Nanowires Fabricated by Top-Down Method. *Adv. Energy Mater.* **7**, 1602138 (2016).
- A2 Lotgering, F. K. Topotactical reactions with ferrimagnetic oxides having hexagonal crystal structures-I. *J. Inorg. Nucl. Chem.* **9**, 113-123 (1959).
- A3 Kobayashi, T., Ogawa, R., Miyazawa, K. i. & Kuwabara, M. Fabrication of β -BaB₂O₄ thin films with (00l) preferred orientation through the chemical solution deposition technique. *J. Mater. Res.* **17**, 844-851 (2011).
- A4 Wang, X. *et al.* Texturing degree boosts thermoelectric performance of silver-doped polycrystalline SnSe. *NPG Asia Mater.* **9**, e426 (2017).

▪ **Reviewer #2**

General comment: In this manuscript, the authors prepared SnSe thin films by solution process, and then conducted characterization for their structure and TE properties. The work is interesting, while the authors should clarify the following questions to improve this work:

Comment 1: The process from SnSe₂ to SnSe cannot be termed with phase transition since they are not allotrope.

Response: We agree with the reviewer's comment that we used the terminology of 'phase transition' in a somewhat broad sense. Accordingly, **we replaced the term of 'phase transition' with 'composition change' or 'transition from SnSe₂ to SnSe'** in the entire manuscript.

Comment 2: The transition from SnSe₂ to SnSe is crucial in this work. The authors should provide more evidence, such as more detailed structural characterization and first-principles calculations, to demonstrate such process other than XRD.

Response: We appreciate the fruitful comment by the reviewer. To further confirm the transition from SnSe₂ to SnSe, firstly, **we conducted Raman spectroscopy analysis** on the samples heated at various temperatures, because the Raman analysis is widely utilized to characterize two-dimensional (2D) metal dichalcogenides. Moreover, the Raman signals of SnSe₂ and SnSe are clearly distinguished in the spectrum. **At 300 °C, the Raman spectrum (Fig. B1) exclusively showed two peaks at 117 cm⁻¹ and 185 cm⁻¹, corresponding to the vibration modes of E_g and A_{g1} of the SnSe₂ phase, respectively^{B1}. On the contrary, at 400 °C, these peaks are totally disappeared and new peaks at 70, 127 and 150 cm⁻¹ and at 107 cm⁻¹ appeared in the Raman spectrum (Fig. B1), which are exactly matched to the B_{3g} and to the A_g vibration modes of the SnSe phase^{B2}.** These results, along with the previous data of the X-ray diffraction (XRD) patterns, the X-ray photoelectron (XPS) spectra, and the energy dispersive spectroscopy (EDS) spectra, clearly evidenced the transition from SnSe₂ to SnSe arising from the Se evaporation.

Fig. B1 | Raman spectra of the SnSe thin films annealed at 300 °C and 400 °C.

Secondly, to understand this transition thermodynamically, we conducted **the first-principle calculation for Helmholtz free energy of both SnSe₂ and SnSe from the density functional theory (DFT)**. The calculations were performed with Projector Augmented Wave (PAW) method^{B3}, Generalized Gradient Approximation (GGA-PBE) exchange-correlation functional^{B4} and DFT-D3 method^{B5} for van der Waals interactions, which are implemented in Vienna *ab initio* simulation package (VASP)^{B6}. The basic unit cell structures for calculation were shown in Fig. B2a (SnSe₂) and b (SnSe). As manifested in the calculated Helmholtz free energy of SnSe₂ and SnSe (Fig. B2c), **SnSe is thermodynamically more stable compared to SnSe₂ in the temperature range from 300 K to 800 K**. Accordingly, SnSe₂ can be predominantly synthesized only in certain circumstance, which can be the Se excess condition. Since the Sn:Se atomic ratio of our purified SnSe ink solution was 1:~2.6, SnSe₂ can be formed as intermediate state, then SnSe₂ transition to SnSe as the heat treatment procedure processed and Se gradually evaporated. These theoretical results further support the transition from SnSe₂ to SnSe during heat treatment in the current study.

Fig. B2 | Thermodynamic free energy calculation. **a**, The unit cell structures for calculation. Supercells for SnSe₂ and SnSe were expanded as much as 4 × 3 × 2 and 2 × 2 × 2, respectively. **b**, Helmholtz free energy of SnSe₂ and SnSe from first-principle calculation.

We included the Raman spectrum (Fig. B1) in the revised Supplementary Information (Supplementary Fig. 5), and included the first-principle calculation results (Fig. B2) in the revised manuscript (Fig. 4). Also, the following discussion and the detailed methods were included in the revised manuscript (page 9-11 and 18-19).

“The Raman spectrum of the thin films annealed at 300 °C exclusively showed two peaks at 117 cm⁻¹ and 185 cm⁻¹, corresponding to the vibration modes of *E_g* and *A_{g1}* of SnSe₂ phase, respectively⁵¹. On the contrary, at 400 °C, these peaks are totally disappeared and new peaks at 70, 127 and 150 cm⁻¹ and at 107 cm⁻¹ appeared in the Raman spectrum (Supplementary Fig. 5), which are exactly matched to the *B_{3g}* and to the *A_g* vibration modes of the SnSe phase⁵².”

“To further understand this transition theoretically, we conducted the first-principle calculation for Helmholtz free energy of both SnSe₂ and SnSe from the density functional theory (DFT). The calculations were performed with Projector Augmented Wave (PAW) method⁵³, Generalized Gradient Approximation (GGA-PBE) exchange-correlation functional⁵⁴ and DFT-D3 method⁵⁵ for van der Waals interactions, which are implemented in Vienna *ab initio* simulation package (VASP)⁵⁶. The basic unit cell structures for calculation were shown in Fig. 4a (SnSe₂) and b (SnSe). As shown in the calculated Helmholtz free energy of SnSe₂ and SnSe (Fig. 4c), SnSe is thermodynamically more stable compared to SnSe₂ in the entire temperature range from 300 K to 800 K. Accordingly, SnSe₂ can be predominantly synthesized only in certain circumstance, which can be the Se excess condition. Since the Sn:Se atomic ratio of our purified SnSe ink solution was 1:~2.6, SnSe₂ can be formed as intermediate state, then SnSe₂ transit to SnSe as the heat treatment procedure processed and Se gradually evaporated. These theoretical results further support the transition from SnSe₂ to SnSe during heat treatment in the current study.”

“Helmholtz free energy calculation. Density functional theory (DFT) calculations were performed with Projector Augmented Wave (PAW) method⁵¹, Generalized Gradient Approximation (GGA-PBE) exchange-correlation functional⁵² and DFT-D3 method⁵³ for van der Waals interactions, which are implemented in Vienna *ab initio* simulation package (VASP)⁵⁴. Valence electrons were set to 4d, 5s and 5p for Sn, 4s and 4p for Se. Fully relaxed crystal structures of SnSe₂ and SnSe are confirmed as *P-3m1* and *Pnma*, respectively. The relaxed lattice constants are 3.858/3.858/6.205 (a/b/c) for SnSe₂ and 11.577/4.171/4.527 for SnSe, respectively. Free energies are post-processed by using phonopy⁶⁷ from DFT calculations of the relaxed structures. Helmholtz free energy (*F*) is calculated based on lattice dynamics⁶⁸ and partition functions (*Z*) of phonon frequencies, where $F = -k_B T \ln Z$. Detail formulations are well-described in developer’s papers^{67,69}.”

Comment 3: The thermal conductivity is key to determine the TE performance, nowadays, there are many works are conducted to obtain the thermal conductivity of thin film by experiments or first-principles calculations. So the authors should give the thermal conductivity to show the TE performance of their SnSe thin film.

Response: We agree with the reviewer’s comment that the thermal conductivity is as much important as other factors to determine the thermoelectric (TE) performance. Typical measurement methods such as 3Ω method or time-domain thermoreflectance (TDTR) for the thermal conductivity of thin films can estimate the properties along the through-plane direction in thin films, which is not appropriate to our textured SnSe thin films exhibiting extremely anisotropic properties. Accordingly,

we conducted the **first-principles DFT calculations for the evaluation of the thermal conductivity of SnSe**. The basic unit cell structure was $\text{Sn}_{16}\text{Se}_{16}$ (Fig. B3a). Moreover, considering the composition of our samples, we further calculate the thermal conductivities of $\text{Sn}_{15}\text{Se}_{16}$ (Fig. B3b) and $\text{Sn}_{16}\text{Se}_{15}$ (Fig. B3c) structures by Debye-Callaway model with the consideration of phonon scattering effect from Umklapp and normal processes, and point defect scattering. **These $\text{Sn}_{15}\text{Se}_{16}$, $\text{Sn}_{16}\text{Se}_{16}$, and $\text{Sn}_{16}\text{Se}_{15}$ are matched to the 5 min-, 9 min-, and 13 min-treated samples in the experiments.** The 1 min-treated sample was excluded for the calculation due to the distinct residual SnSe_2 , causing the large deformation in the simulated unit cell structure.

As shown in Fig. B3d, the calculation result with unit cell having atomic ratio 16:16 has the highest thermal conductivity of $1.53 \text{ W m}^{-1} \text{ K}^{-1}$ at room temperature, while $\text{Sn}_{15}\text{Se}_{16}$ and $\text{Sn}_{16}\text{Se}_{15}$ shows the values of $0.82 \text{ W m}^{-1} \text{ K}^{-1}$ and $0.89 \text{ W m}^{-1} \text{ K}^{-1}$, respectively. The lower thermal conductivities are attributed to Sn or Se vacancies acting as scattering center. With these calculated thermal conductivities and newly measured electrical properties at high temperatures (Fig. B5), we estimated the ZT values of our samples. **The further detailed discussion about the TE properties is described in the Response for the Comment 4.**

Fig. B3 | First-principle calculation for thermal conductivities of SnSe. The unit cell structures of (a) $\text{Sn}_{16}\text{Se}_{16}$, (b) $\text{Sn}_{15}\text{Se}_{16}$, and (c) $\text{Sn}_{16}\text{Se}_{15}$ for calculation. Gray, green, and orange spheres in a-c represents Sn, Se, and vacancy, respectively. **d**, Calculated thermal conductivities for SnSe with Sn:Se atomic ratio of 15:16, 16:16, and 16:15.

We included the calculated thermal conductivities (Fig. B3) in the revised Supplementary Information (Supplementary Fig. 9) and included the following detailed methods in the revised manuscript (page 19-20).

“Thermal conductivity calculation. In $\text{Sn}_{16}\text{Se}_{16}$ case, the lattice thermal conductivity was computed from a solution of the linearized phonon Boltzmann transport equation (LBTE)⁷⁰. In single mode relaxation time (SMRT) approximation, the equation can be written in a closed form,

$$\kappa = \frac{1}{NV_0} \sum_{\lambda} C_{\lambda} \mathbf{v}_{\lambda} \otimes \mathbf{v}_{\lambda} \tau_{\lambda}^{\text{SMRT}}$$

where V_0 is the volume of a unit cell, and \mathbf{v}_{λ} and $\tau_{\lambda}^{\text{SMRT}}$ are the group velocity and single mode relaxation time (SMRT) of phonon mode λ , respectively.

In order to obtain phonon dispersion relation and phonon lifetimes, PHONOPY⁶⁷, and PHONO3PY⁷⁰ code were employed. Harmonic and anharmonic interatomic force constants (IFCs) were calculated using VASP^{71,72} with the PBEsol⁷³ exchange–correlation functional. The atomic displacement was set to 0.03 Å, and the k-meshes for the force calculations were set to $2 \times 2 \times 2$. To calculate the lattice thermal conductivity, the q-meshes were set to $7 \times 7 \times 7$.

In $\text{Sn}_{16}\text{Se}_{15}$ and $\text{Sn}_{15}\text{Se}_{16}$ case, lattice thermal conductivity calculations were performed using the Callaway model, which was tuned to match with Boltzmann Transport Equation (BTE) results, suggested by D. Wu *et al.*⁷⁴ Three scattering factors were considered: the intrinsic Umklapp and Normal processes, and point defect scattering. In the relaxation time approximation, the lattice thermal conductivity κ_{latt} can be written as:

$$\kappa_{\text{latt}} = \frac{k_B}{2\pi^2 v} \left(\frac{k_B T}{\hbar} \right)^3 \int_0^{\theta_D/T} \frac{\tau_c z^4 e^z}{(e^z - 1)^2} dz$$

the total relaxation time (τ_c) consists of individual scattering mechanism *via Matthiessen's rule*:

$$\tau_c^{-1} = \tau_U^{-1} + \tau_N^{-1} + \tau_v^{-1}$$

Umklapp process,

$$\tau_U^{-1} = \frac{\hbar \gamma^2}{M v^2 \theta_D} \omega^2 T \exp\left(-\frac{\theta_D}{3T}\right)$$

Normal process,

$$\tau_N^{-1} \approx \beta \tau_U^{-1}$$

Vacancy scattering,

$$\tau_v^{-1} = \frac{\omega^4 \delta^3}{4\pi v^3} \gamma (1 - \gamma) \left[-\frac{M_v}{M} - 2 \right]^2$$

where k_B is the Boltzmann constant, \hbar is the Plank constant, v is the sound (phonon-group) velocity along each principle axis at the long wave limit, θ_D is the axial Debye temperature, and z is defined as $\hbar\omega/k_B T$, γ is grüneisen parameter, M is average molar mass of one atom; β is a fitting parameter for

Normal process; δ is the volume of a host atom; M_v is the molar mass of the missing atom (vacancy), y is the molar ratio of Sn vacancies;”

Comment 4: According to Zhao's work of Nature (Ref. 14), the peak of ZT is at 923 K. The highest temperature given in the present manuscript is 473 K, I think the authors should perform TE characterization at higher temperatures.

Response: We fully agree with the reviewer that the high temperature measurement of TE properties the current textured SnSe thin films is crucial to compete with the previously reported SnSe bulk. Since the slight changes of the atomic composition in the SnSe thin film can cause the huge degradation of the TE properties, it is crucial to prevent from the evaporation of Se from the SnSe thin films during the measurement at high temperature. The previously measured results obtained from the SnSe thin films showed the abrupt drop of electrical conductivity above 500 K (Fig. B4). This signify the SnSe thin film underwent the unwanted compositional change due to the evaporation of Se during the measurement.

Fig. B4 | Temperature dependent electrical conductivity of 13 min-treated sample without oxide layer.

Accordingly, we tried to deposit the 35 nm-thick Al_2O_3 thin protecting layer onto the SnSe thin films using the atomic layer deposition (ALD) technique. Then, we measured the electrical conductivity and the Seebeck coefficient of these samples at temperatures ranging from 300 K to 750 K. During the measurement, we didn't observe any breakdown related to the Se evaporation in the electrical properties at all, which indicates that the Al_2O_3 layer successfully prevent the samples from the Se evaporation. To the best of our knowledge, such high-temperature TE properties have never been reported on TE metal chalcogenides thin films like SnSe because

metal chalcogenides thin films usually have a **critical issue of thermal stability** arising from the evaporation of chalcogens.

The electrical conductivities of all samples exhibited a negative temperature dependence in temperature below ~ 650 K (Fig. B5a), then change to semiconducting behavior for the higher temperature region, which agrees with the behavior observed in SnSe single crystals with the *Pnma* space group^{B7}. Owing to the relationship of hole concentrations and hole mobility with heat treatment, the electrical conductivity decreased with increasing heat treatment time. The highest electrical conductivity of 7.6×10^3 S cm⁻¹ at 750 K was achieved by the 1 min-heated sample; this value is similar to or even higher than those reported for SnSe single crystals and polycrystals. The Seebeck coefficients of the samples were between 200~300 μ V K⁻¹ at room temperature but increased with heat treatment time because Seebeck coefficients are inversely proportional to carrier concentrations. The 1 min- and 5 min-treated samples showed significant decrease in Seebeck coefficients as temperatures increased. This temperature dependence can be attributed to the residual SnSe₂ phase in less heat-treated samples. Typically, 2D SnSe₂ crystals are reported to exhibit n-type semiconductor properties^{B8}, in which the thermally excited minority carrier concentration can be increased at high temperature and it significantly suppress the Seebeck coefficient in the current p-type SnSe thin films. On the other hand, 9 min- and 13 min-treated samples showed similar behavior to the behaviors of typical SnSe single crystals and polycrystals. **The highest power factors achieved by the 9 min-sample at 550 K was 4.27 μ W cm⁻¹ K⁻². To the best of our knowledge, this value is at least an order of magnitude higher than the most recently reported SnSe thin films^{B9} and even higher than those measured along the *b*-axis of SnSe single crystals in the mid-temperature range^{B7}.**

Fig. B5 | Temperature dependent TE properties of SnSe thin films. a-d, Graphs show electrical conductivities (a), Seebeck coefficients (b), power factors (c), ZT values (d) for SnSe thin films annealed at 400 °C for 1 min, 5 min, 9 min, and 13 min. ZT values were estimated with the calculated thermal conductivities of $\text{Sn}_{15}\text{Se}_{16}$, $\text{Sn}_{16}\text{Se}_{16}$, and $\text{Sn}_{16}\text{Se}_{15}$ structures, which matched with the compositions of 5 min-, 9 min- and 13 min-treated samples. Stars in (c) and (d) indicate data points obtained with SnSe single crystals in a-(black), b-(red), and c-(blue) axes^{B7}.

With the calculated thermal conductivities of $\text{Sn}_{15}\text{Se}_{16}$, $\text{Sn}_{16}\text{Se}_{16}$, and $\text{Sn}_{16}\text{Se}_{15}$, we estimated the ZT values of 5 min-, 9 min- and 13 min-treated samples. **The maximum ZT value of 0.58 was achieved by 13 min-treated sample at 750 K. To the best of our knowledge, this value is the highest among the reported of SnSe thin films and compete with those measured along the *b*-axis of SnSe single crystals in the corresponding temperature range.** Also, considering the similar temperature dependence of electric properties with single crystalline bulk SnSe, we speculate that power factors of our samples will increase after 750 K. Moreover, it is noteworthy that the thin films generally have lower thermal conductivity than bulk sample, originated from surface phonon scattering effect. We expect calculated thermal conductivities presented on Fig. B3d is slightly overestimated value, which could devalue the actual ZT value.

We replaced the previously measured temperature dependent TE properties of SnSe thin films (Fig. B5a-c) in the revised manuscript (Fig.6), and included the following discussion included in

the revised manuscript (page 13-15). The estimated ZT values of our samples (Fig. B5d) were included in the revised Supplementary Information. (Supplementary Fig. 10)

“Temperature dependent TE properties were measured at temperatures ranging from room temperature to 750 K. The electrical conductivities of all samples exhibited a negative temperature dependence in temperature below ~650 K (Fig. 6a), which was then changed to semiconducting behaviour for the higher temperature region, agreeing with the behaviour observed in SnSe single crystals with the *Pnma* space group¹⁴. Owing to the relationship of hole concentrations and hole mobility with heat treatment, the electrical conductivity decreased with increasing heat treatment time. The highest electrical conductivity of $7.6 \times 10^3 \text{ S cm}^{-1}$ at 750 K was achieved by the 1 min-heated sample; this value is similar to or even higher than those reported for SnSe single crystals and polycrystals. The Seebeck coefficients of the samples were between 200~300 $\mu\text{V K}^{-1}$ at room temperature but increased with heat treatment time because Seebeck coefficients are inversely proportional to carrier concentrations. The 1 min- and 5 min-treated samples showed significant decrease in Seebeck coefficients as temperatures increased. This temperature dependence can be attributed to the residual SnSe₂ phase in less heat-treated samples. Typically, 2D SnSe₂ crystals are reported to exhibit n-type semiconductor properties^{46,64,65}, in which the thermally excited minority carrier concentration can be increased at high temperature and it significantly suppress the Seebeck coefficient in the current p-type SnSe thin films. On the other hand, 9 min- and 13 min-treated samples showed similar behaviour to typical SnSe single crystals and polycrystals. The highest power factors achieved by the 9 min-sample at 550 K was $4.27 \mu\text{W cm}^{-1} \text{ K}^{-2}$. To the best of our knowledge, this value is at least an order of magnitude higher than the most recently reported SnSe thin films³³ and even higher than those measured along the *b*-axis of SnSe single crystals in the mid-temperature range¹⁴.

The thermal conductivities of the SnSe thin films were estimated by the first-principles DFT calculations because accurate measurements of in-plane thermal conductivity are technically difficult to be obtained by typical methods such as 3 Ω method or time-domain thermoreflectance (TDTR). The basic unit cell structure for the calculation was Sn₁₆Se₁₆ (Supplementary Fig. 9a). Moreover, considering the composition of our samples, we further calculated the thermal conductivities of Sn₁₅Se₁₆ and Sn₁₆Se₁₅ structures by Debye-Callaway model with the consideration of phonon scattering effect from Umklapp and normal processes, and point defect scattering. These Sn₁₅Se₁₆, Sn₁₆Se₁₆, and Sn₁₆Se₁₅ are generally matched to the fabricated 5 min-, 9 min-, and 13 min-treated samples in the composition. The 1 min-treated sample was excluded for the calculation due to the distinct residual SnSe₂, causing a large deformation in the simulated unit cell structure. As shown in the Supplementary Fig. 9d, the calculation result with unit cell having atomic ratio 16:16 has the highest thermal conductivity of $1.53 \text{ W m}^{-1} \text{ K}^{-1}$ at room temperature, while Sn₁₅Se₁₆ and Sn₁₆Se₁₅

shows the values of $0.82 \text{ W m}^{-1} \text{ K}^{-1}$ and $0.89 \text{ W m}^{-1} \text{ K}^{-1}$, respectively. The lower thermal conductivities of $\text{Sn}_{15}\text{Se}_{16}$ and $\text{Sn}_{16}\text{Se}_{15}$ are attributed to Sn or Se vacancies acting as scattering centres. Moreover, all models showed the negative temperature dependence of thermal conductivities, agreeing with the typical trend of semiconductors.

With the calculated thermal conductivities of $\text{Sn}_{15}\text{Se}_{16}$, $\text{Sn}_{16}\text{Se}_{16}$, and $\text{Sn}_{16}\text{Se}_{15}$, we estimated the ZT values of 5 min-, 9 min- and 13 min-treated samples. The maximum ZT value of 0.58 was achieved by 13 min-treated sample at 750 K. To the best of our knowledge, this value is the highest among the reported of SnSe thin films and compete with those measured along the *b*-axis of SnSe single crystals in the corresponding temperature range (Supplementary Fig. 10). Also, considering the similar temperature dependence of electrical properties with single crystalline bulk SnSe, the power factors of our samples may be further increased after 750 K. Considering that thin films typically have much lower thermal conductivity than bulk materials owing to interface or surface scattering of phonons, the actual ZT value of the current SnSe thin films is expected to be higher than the calculated.”

References

- B1 Fernandes, P. A., Sousa, M. G., Salomé, M. P., Leitão, J. P. & da Cunha, A. F. Thermodynamic pathway for the formation of SnSe and SnSe₂ polycrystalline thin films by selenization of metal precursors. *Cryst. Eng. Comm.* **15**, 01278 (2013).
- B2 Chandrasekhar, H. R., Humphreys, R. G., Zwick, U. & Cardona, M. Infrared and Raman spectra of the IV-VI compounds SnS and SnSe. *Phys. Rev. B* **15**, 2177 (1977).
- B3 Kresse, G. From ultrasoft pseudopotentials to the projector augmented-wave method. *Phys. Rev. B* **59**, 1758-1775 (1999).
- B4 Perdew, J. P., Burke, K. & Ernzerhof, M. Generalized gradient approximation made simple. *Phys Rev. Lett.* **77**, 3865-3868 (1996).
- B5 Grimme, S., Antony, J., Ehrlich, S. & Krieg, H. A consistent and accurate *ab initio* parameterization of density functional dispersion correction (DFT-D) for the 94 elements H-Pu. *J. Chem. Phys.* **132**, 154103 (2010).
- B6 Kresse, G. & Hafner, J. *Ab initio* molecular dynamics for liquid metals. *Phys. Rev. B* **47**, 558-561 (1993).
- B7 Zhao, L. D. et al. Ultralow thermal conductivity and high thermoelectric figure of merit in SnSe crystals. *Nature* **508**, 373-377 (2014).
- B8 Mitzi, D. B., Kosbar, L. L., Murray, C. E., Copel, M. & Afzali, A. High-mobility ultrathin semiconducting films prepared by spin coating. *Nature* **428**, 299-303 (2004).
- B9 Burton, M. R. et al. Thin film tin selenide (SnSe) thermoelectric generators exhibiting ultralow thermal conductivity. *Adv. Mater.* **30**, e1801357 (2018).

▪ **Reviewer #3**

General comment: This paper “Phase transition-induced texturing and doping in solution-processed SnSe thermoelectric thin films” by Heo et al. reports the high thermoelectric performance of solution-processed SnSe thin film. The detailed synthetic chemistry for ink solution was shown to obtain molecular Sn₂Se₆ chalcogenidometallate complex. The most interesting part of this paper is the phase transition from SnSe₂ to SnSe and the resulting texturing in thin films, which also seems to be beneficial to control the composition and the related doping. The achieved power factor of ~3 μW cm⁻¹ K⁻² by SnSe thin films is a really high value in thin film thermoelectrics and comparable to bulk SnSe. Although the strategies of texturing and doping in bulk SnSe have been studied in several papers to enhance the thermoelectric conversion efficiency, I think that the phase-transition approach for this material is new and can offer an important scientific insight to thermoelectric society. Moreover, the paper is well organized and the results and discussion including experimental and theoretical considerations is reasonable. In these regards, I recommend the publication of this paper in Nature Communications after addressing some minor issues, as presented below.

Comment 1: In XPS and XRD analyses of SnSe thin films, the substantial portion of SnSe₂ phase was detected in 1 min sample. I think this residual phase can significantly affect the carrier density and mobility. For example, the authors already discussed n-type properties of SnSe₂ phase, affecting the thermoelectric properties of SnSe thin films in the manuscript. The authors need to discuss this point.

Response: We appreciate the valuable comment by the reviewer and agree that the residual SnSe₂ phase can affect the electrical properties in the fabricated SnSe thin film. For example, very recently, Wang, et al. reported that locally embedded SnSe₂ microdomains in the SnSe phase lead to the hole doping in SnSe through the interfacial charge transfer^{C1}. In our case, broad and subtle SnSe₂ (001) peak was detected in the 1 min-treated sample, which is considered as a trace of residual SnSe₂ phase during the transition from SnSe₂ to SnSe. Since any types of the peak shift related to alloying or structural deformation in X-ray diffraction (XRD) pattern of the 1 min-treated sample were not observed at all, this result suggests that the residual SnSe₂ phase exists as microdomains within the SnSe phase, contributing to hole doping.

Accordingly, we included the following sentences in the revised manuscript (page 13).

“At the same time, we cannot exclude the possibility that the residual SnSe₂ phase contributed to the overall electrical properties in less-heat-treated thin films. Recently, Wang, et al. reported that locally embedded SnSe₂ microdomains in the SnSe phase lead to the hole doping in SnSe through the

interfacial charge transfer⁶³. Likewise, the electrical properties of the 1 min-treated SnSe sample might also be affected by the residual SnSe₂ phase existing as microdomains within the SnSe phase.”

Comment 2: In 30 min sample, the author commented “the continuous microstructures deformed severely.” However, the microstructure seen in SEM image is not so different from other samples and a new phase of rod structure was observed. I recommend the authors provide more detailed information of this sample.

Response: As the reviewer suggested, to further understand the structural deformation occurring at the 30 min-treated SnSe thin film, we carried out the microstructural analysis using the scanning electron microscopy (SEM) and the energy dispersive X-ray spectroscopy (EDS) analyses. As shown in the SEM image (Figure C1a), the plate-like textured microstructures were partially deformed and the perpendicularly oriented large-sized plate newly appeared. The elemental mapping image of this sample (Figure C1b) reveals that Sn and Se are homogeneously distributed over the entire area without the local accumulation, indicating the composition of large-sized plates is identical to that of the matrix. However, the composition of this sample estimated by EDS analysis is only 1:0.81 of the atomic ratio of Sn:Se, where the Se content is much lower than those observed in other samples (1 min~13 min heat-treated). In fact, such a Se deficiency critically affected the electrical properties; e.g. the electrical conductivity of this sample is at least an order of magnitude lower than other samples. Accordingly, we focused on the samples heated for less than 15 min for the characterization of thermoelectric properties.

Fig. C1 | (a) SEM image of SnSe thin film heat treated over 30 min and (b) EDS mapping image of the area shown in Figure C1a.

We replaced the previous SEM image with the newly analyzed. SEM and EDS mapping image

(Figure C1) in the revised Supplementary Information (Supplementary Fig. 8) and included the following sentences in the revised manuscript (page 11-12).

“In contrast, when the thin film was annealed longer than 30 min, the plate-like textured microstructures were partially deformed while the perpendicularly oriented large-sized plate were newly appeared (Supplementary Fig. 8a). The elemental mapping image of this sample (Supplementary Fig. 8b) reveals that Sn and Se are homogeneously distributed over the entire area without the local accumulation, indicating the composition of large-sized plates is identical to that of the matrix. However, the composition of this sample estimated by EDS analysis is only 1:0.81 of the atomic ratio of Sn:Se, where the Se content is much lower than those observed in other samples (1 min~13 min heat-treated). In fact, such a Se deficiency critically affected the electrical properties; e.g. the electrical conductivity of this sample is at least an order of magnitude lower than other samples.”

References

- C1 Wang, Z. *et al.* Defects controlled hole doping and multivalley transport in SnSe single crystals. *Nat. Commun.* **9**, 47 (2018).

REVIEWERS' COMMENTS:

Reviewer #1 (Remarks to the Author):

The authors have done a thorough job of addressing reviewer concerns with the original manuscript. In particular, I believe the addition of data for the spray-deposited films and the addition of measurements at higher temperature significantly strengthen the manuscript. I recommend publication

Reviewer #3 (Remarks to the Author):

The raised issues in the first review have been adequately addressed. Thus I recommend "publish as is".